# A pathogen-induced putative NAC transcription factor mediates leaf rust resistance in barley

Chunhong Chen [1,6], Matthias Jost[1,6], Megan A. Outram[1], Dorian Friendship[2], Jian Chen [1], Aihua Wang[1], Sambasivam Periyannan[1,3], Jan Bartoš [4], Kateřina Holušová[4], Jaroslav Doležel [4], Peng Zhang [2], Dhara Bhatt[1], Davinder Singh [2], Evans Lagudah [1] ✉, Robert F. Park [2] ✉ & Peter M. Dracatos [2,5,6] ✉

Leaf rust, caused by *Puccinia hordei*, is one of the most widespread and damaging foliar diseases affecting barley. The barley leaf rust resistance locus *Rph7* has been shown to have unusually high sequence and haplotype divergence. In this study, we isolate the *Rph7* gene using a fine mapping and RNA-Seq approach that is confirmed by mutational analysis and transgenic complementation. *Rph7* is a pathogen-induced, non-canonical resistance gene encoding a protein that is distinct from other known plant disease resistance proteins in the Triticeae. Structural analysis using an AlphaFold2 protein model suggests that *Rph7* encodes a putative NAC transcription factor with a zinc-finger BED domain with structural similarity to the N-terminal DNA-binding domain of the NAC transcription factor (ANAC019) from *Arabidopsis*. A global gene expression analysis suggests *Rph7* mediates the activation and strength of the basal defence response. The isolation of *Rph7* highlights the diversification of resistance mechanisms available for engineering disease control in crops.

Cultivated barley (*Hordeum vulgare* L.) is the world's fourth most important cereal, used primarily in malt production for alcoholic beverages and as grain feed for livestock and human food[1]. Of concern though are foliar diseases that reduce yield, grain quality and profitability[2]. Leaf rust, caused by *Puccinia hordei* Otth., is widespread, resulting in regular seasonal epidemics and economically significant losses in global barley production[3]. Resistance to leaf rust in *Hordeum* spp. is well characterised and widely available due to the numerous genetic[3–12] and cloning studies undertaken[13–16]. However, in most cases the underlying genes responsible for resistance are not

known, limiting effective and efficient deployment in agricultural settings[3,17].

The plant immune system is multi-layered, with both extra- and intracellular receptors being critical for pathogen recognition and disease resistance signalling[18]. Race-specific resistance, referring to defence against certain races of a pathogen, occurs via perception of the pathogen governed by two predominant immune receptor families: nucleotide binding-leucine-rich repeat (NLRs) and receptor-like kinase (RLKs) receptors. To date, the most prevalent immune receptors characterised in cereal crops are NLRs. However, recent evidence

[1]CSIRO Agriculture and Food, Commonwealth Scientific and Industrial Research Organisation, GPO Box 1700, Canberra, ACT 2601, Australia. [2]The University of Sydney, Faculty of Science, Plant Breeding Institute, Cobbitty, NSW 2570, Australia. [3]The University of Southern Queensland, School of Agriculture and Environmental Science, Centre for Crop Health, Toowoomba, QLD 4350, Australia. [4]Institute of Experimental Botany, Centre of Plant Structural and Functional Genomics, Olomouc CZ-77900, Czech Republic. [5]La Trobe Institute for Sustainable Agriculture & Food (LISAF), Department of Animal, Plant and Soil Sciences, La Trobe University, Melbourne, VIC 3086, Australia. [6]These authors contributed equally: Chunhong Chen, Matthias Jost, Peter M. Dracatos. ✉e-mail: evans.lagudah@csiro.com.au; robert.park@sydney.edu.au; p.dracatos@latrobe.edu.au

suggests race-specific resistance is also governed by non-canonical gene classes[17,19,20]. Two recent studies utilised sequenced chromosome scale assemblies from the wheat and barley pangenomes to clone the race-specific leaf rust resistance genes *Lr14a*[21] and *Rph3*[16]. Interestingly, *Lr14a* and *Rph3* encode a membrane-bound ankyrin repeat containing protein and a putative executor protein, respectively. These studies highlight opportunities to both explore and exploit diverse resistance mechanisms in cereal crops.

*Rph7* is a semi-dominant inherited gene, first described from the cultivar 'Cebada Capa' (PI 539113)[22]. *Rph7* confers all-stage resistance against most *P. hordei* isolates in Europe and Australia, though isolates virulent on *Rph7* have been identified in Spain[23], the Near East[24], North America[25], and most recently in Australia. Trisomic analysis initially mapped *Rph7* to barley chromosome 3H[26] and subsequent biparental mapping further refined the *Rph7* locus to the short arm of chromosome 3H, near the telomere[27]. Fine mapping and sequencing of physically overlapping BAC clones spanning the determined interval revealed that *Rph7* is located within a chromosomal region of high haplotypic divergence, largely explained by the presence of numerous cultivar-specific insertions containing different classes of retrotransposons[28,29]. Shotgun Sanger sequencing and assembling of BAC clones of the leaf rust susceptible cultivar Morex and the *Rph7*-containing line Cebada Capa revealed the presence of a 100 kb sequence insertion in Cebada Capa while the genes at the *Rph7* locus were conserved[29]. No typical resistance gene candidates were identified within the cultivar-specific insertion. Therefore, four co-segregating genes (*HvHGA1*, *HvHGA2*, *HvPG1* and *HvPG4*) were considered as the most logical candidates for the *Rph7* resistance. Despite an in-depth comparative sequence analysis and haplotypic characterisation at the *Rph7* locus, the causal gene underpinning the resistance was not resolved.

In the present study, we build on the evidence of Scherrer et al.[29] with the aim of determining the molecular genetic basis of the leaf rust resistance gene *Rph7* resistance in barley donor cultivar Cebada Capa. We report on cloning the *Rph7* gene from barley by performing additional recombination-based mapping, RNA-Seq expression analysis and mutagenesis. RNA-Seq data shows that *Rph7* modulates the basal defence response and similar to the recently cloned leaf rust resistance gene *Rph3*, *Rph7* is only expressed following challenge with *P. hordei* pathotypes that carry the corresponding *Avr* gene. *Rph7* induces partial cell death in both *Nicotiana benthamiana* and barley protoplasts but lacks the transmembrane domain characteristic of previously reported executor genes. AlphaFold structural analysis indicates *Rph7* encodes a NAC transcription factor with zinc-finger BED domain at the C terminal that is similar in structure to the NAC transcription factor from *Arabidopsis* (ANAC019).

## Results and discussion
### Identifying gene candidates at the *Rph7* locus
Recent crop pangenome projects have revealed both the extent and importance of intraspecies polymorphisms, including presence-absence variations (PAVs), highlighting the inadequacy of previous over-reliance on reference genome information[30]. PAVs are especially relevant for resistance gene classes that evolve via duplication and diversifying selection (like NLRs). Often the causal resistance gene is either absent or partially represented in susceptible accessions[31]. To resolve the underlying molecular basis of the *Rph7* resistance we tested two hypotheses. The first was that the involvement of one or more of the previously postulated candidate genes (*HvPG1*, *HvPG4*, *HvHGA1* and *HvHGA2*) could be confirmed or eliminated through additional recombinant screening at the *Rph7* locus. The second was that an RNA-Seq based expression analysis would reveal additional candidate genes at the *Rph7* locus. We therefore performed additional genetic fine mapping, using an alternative susceptible barley genotype Wabar2722 (*rph7*) crossed with the *Rph7* donor line Cebada Capa (*Rph7*), to test whether we could identify further recombinants in the previously

defined genetic interval. The progeny-tested recombinants eliminated the involvement of three of the previous *Rph7* gene candidates (*HvHGA1*, *HvHGA2* and *HvPG4*) (Fig. 1a, Supplementary Table 1).

We performed an RNA-Seq experiment at the seedling stage at both early (24 h) and late (day 6) timepoints to determine the genes specifically expressed during infection when challenged with an *Rph7*-avirulent *P. hordei* pathotype. We identified five expressed genes within the target interval, four of which were not predicted in the previous *Rph7* study[29]. Interestingly, three of the genes (viz. *UnkP-1*, *ZnF-BED1* and *UnkP-2*) were upregulated only in the *Rph7*-carrying lines Cebada Capa and BW758 (near isogenic line carrying *Rph7* in the susceptible cv Bowman background) (Fig. 1b, c, Supplementary Table 2). This suggests one or more of these genes may mediate *Rph7* resistance in Cebada Capa. To further verify the involvement of the expressed candidate genes at the locus we chemically mutagenized the BW758 line with the aim of identifying multiple independent *Rph7* knockout mutants for comparative sequence analysis. Ten M$_2$ families were identified as putative *Rph7* knockout mutants. Subsequent progeny testing of M$_3$ families confirmed seven homozygous susceptible and one segregating family for further sequence analysis. Sanger sequencing of the five expressed genes within the target *Rph7* interval (*UnkP-1*, *ZnF-BED1*, *UnkP-2*, *ZnF-BED2* and *HvPG1*) on all progeny-tested susceptible mutants determined that four out of eight mutant lines contained chemically induced SNPs within the coding region (either C > T or G > A) of the *ZnF-BED1* gene (Fig. 2a). No non-synonymous mutations were identified in the remaining four mutant lines in *ZnF-BED1* or in any of the sequenced neighbouring genes. Due to the lack of a continuous sequence at the resistant haplotype, we opted to perform a MutChromSeq experiment to add a second layer of confidence and avoid missing further putative candidates. MutChromSeq is an unbiased approach in complexity reduction used to rapidly isolate plant genes and regulatory DNA sequences that is not reliant on recombination-based genetic mapping[13]. We flow-sorted and sequenced the 3H chromosomes of wild type and seven of the mutant lines and performed MutChromSeq analysis. MutChromSeq analysis confirmed *ZnF-BED1* as the primary candidate and confirmed no additional plausible candidates on chromosome 3H. This suggests the possible presence of mutations in downstream targets of *Rph7* or genes regulating the expression of resistance.

### Candidate gene validation and haplotypic characterisation at the *Rph7* locus
The genomic sequence structure of *ZnF-BED1* consists of 1197 nucleotides, including four exons and three introns, which encode for a 302-residue protein with a sequence-predicted zinc-finger BED domain (ZnF-BED) at the C-terminus (Fig. 2c). To conclusively confirm the involvement of *ZnF-BED1* in mediating *Rph7* resistance we performed a complementation experiment by cloning a 3625 bp genomic fragment including the native promoter and terminator into Golden Promise using *Agrobacterium*-mediated transformation. Rust testing was performed on T$_1$ generation Golden Promise + *ZnF-BED1* lines and controls Golden Promise, Bowman, BW758 and Cebada Capa using four *P. hordei* pathotypes eliciting differential responses on *Rph7*-carrying lines (Fig. 1d, Supplementary Tables 3 and 4). Four T$_1$ lines segregated for a single gene (3 Resistant:1 Susceptible) for the expected *Rph7* phenotype in response to three *Rph7*-avirulent pathotypes. In contrast, the same four T$_1$ lines were susceptible to the *Rph7*-virulent pathotype as the resistant controls confirming the *Rph7* specificity of the Golden Promise transgenic lines. Further genotypic analysis using PCR markers designed to the selectable marker and *ZnF-BED1* was performed to confirm the transgenic status of the lines (Supplementary Table 4). The correlation of segregation patterns observed in the T$_1$ lines (B114-1, B114-2, B114-17, and B114-18) in response to all *P. hordei* pathotypes suggests that a single resistance factor (*ZnF-BED1*) is sufficient to confer *Rph7*-mediated resistance. We

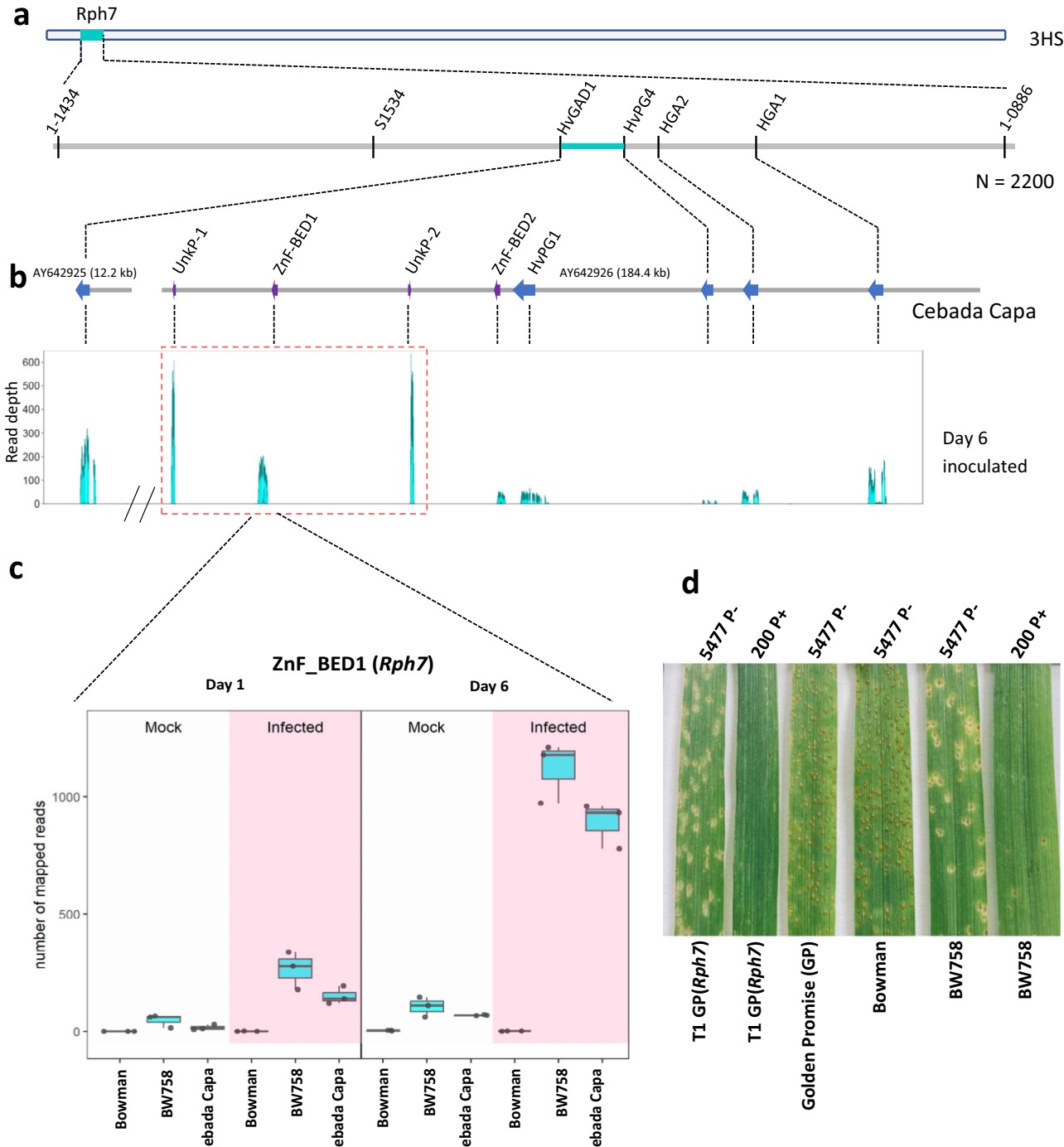

**Fig. 1 | Gene discovery at the *Rph7* locus in Cebada Capa. a** Genetic mapping narrowed the *Rph7* locus between flanking markers within genes *HvGAD1* and *HvPG4*. **b** Schematical overview of previous sequenced BAC contigs[29] (grey). RNA-Seq data revealed the gene expression of previous reported candidate genes (blue) and identified a further four expressed candidate genes labelled in purple. Among the four identified expressed candidates, three genes are upregulated after infection (highlighted in red dashed box) in the resistant lines Cebada Capa and the near isogenic line BW758 (Bowman+*Rph7*) compared to susceptible Bowman at day 6. **c** Expression profile of *ZnF-BED1* (*Rph7*). Samples labelled with white background were treated as mock control, while pink background labelled shows the expression patterns are from barley leaves infected with *Rph7*-avirulent *Puccinia hordei*

pathotype 5457 P+ for Bowman, BW758 and Cebada Capa. The data presented in all cases captures biological replication (*n* = 3) where the box encompasses two middle quartiles, with central line showing the mean. Whiskers extend to the furthest data point within 1.5 times the interquartile range. **d** Complementation results showing the same phenotypic responses between the T$_1$ transgenic Golden Promise + *Rph7* line B114-1 and the *Rph7*-carrying positive control BW758 at the seedling stage 10 days after infection with *Puccinia hordei* pathotypes 5477 P− and 200 P+ that elicit intermediate and low infection types respectively. Leaf rust susceptible controls Golden Promise and Bowman were also inoculated with 5477 P− and demonstrate the *Rph7* specificity. Source data are provided as a Source Data file.

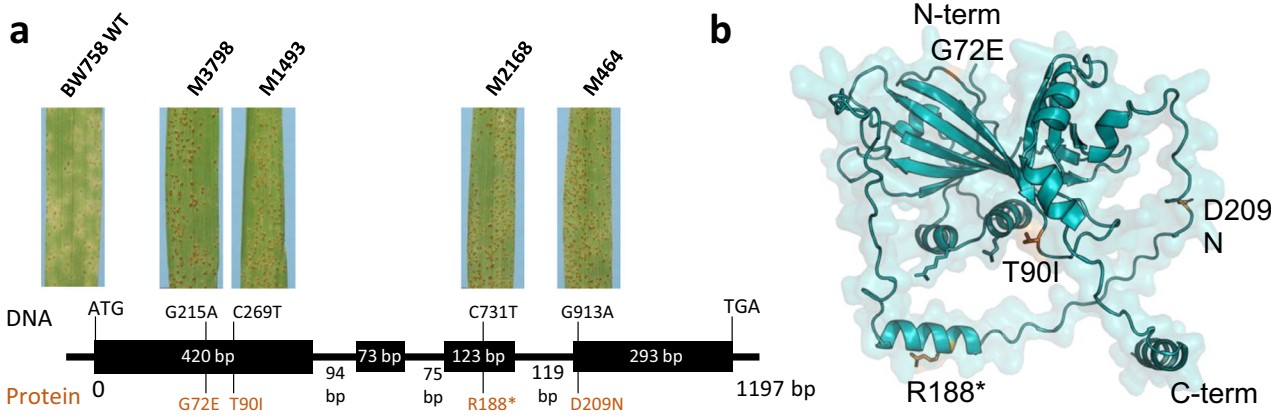

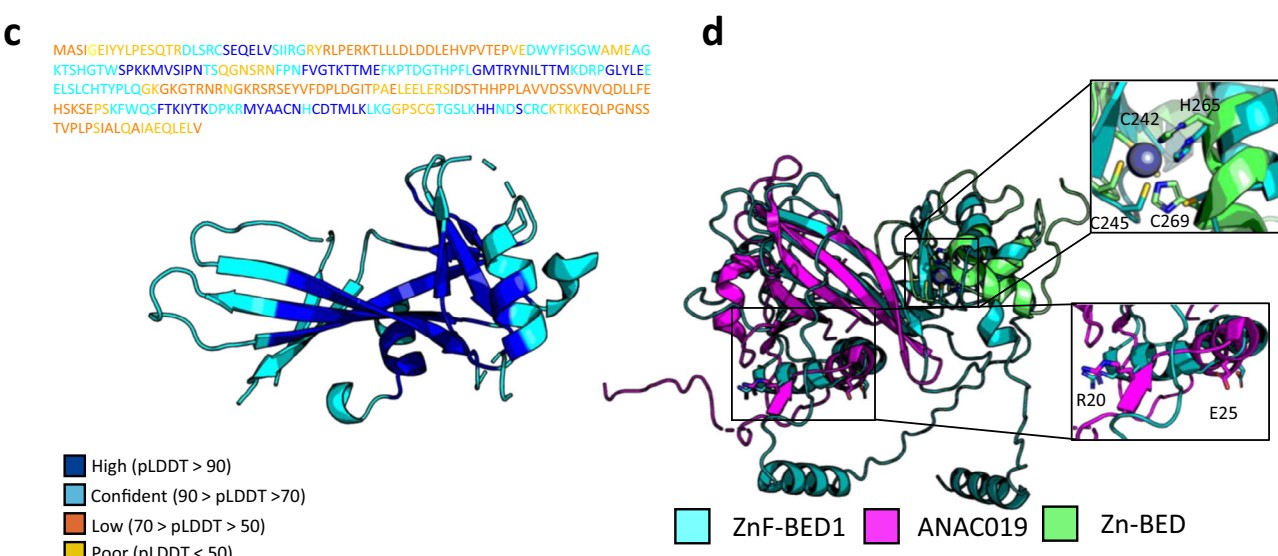

**Fig. 2 | ZnF-BED1 shares structural similarity with plant specific NAC domain-containing proteins. a** Schematical drawing of the gene structure of *ZnF-BED1* consisting of 4 exons (black boxes) and 3 introns with respective length in bp. Phenotypic responses at the seedling stage 10 days after infection with *Rph7*-avirulent *Puccinia hordei* pathotype 5457 P+ of resistant wild type (WT) (BW758) and four independent sodium azide-induced mutants are shown for each of the mutant lines. Positions of four independent mutations with corresponding changes on DNA level (black) and protein level (orange) within the coding sequence indicating that the *ZnF-BED1* gene was required for *Rph7*-mediated resistance. **b** Transparent surface representation of ZnF-BED1 highlighting the four non-synonymous amino acid changes identified in ZnF-BED1 mutant lines. Residues are shown in stick representation and coloured in orange. **c** AlphaFold2 prediction of ZnF-BED1 shown in

cartoon representation. The predicted model is coloured by the per-residue confidence (pLDDT) score bands as depicted, with only confident regions (pLDDT > 70) shown. The sequence above the prediction is coloured according to confidence bands as represented. The full prediction can be found in Fig. S3a. **d** Structural superimposition of the AlphaFold2 prediction of ZnF-BED1 (teal) and top structural match from Dali ANAC019 NAC domain (PDB ID: 3SWP; magenta) shows similarity is limited to the N-terminus of ZnF-BED1, and superimposition of the top structural match from Dali for the C-terminal domain (residues 220-302), the C2H2-type zinc-finger domain of human zinc-finger BED domain-containing protein 2 (PDB ID: 2DJR, green). Insert shows residues involved in Zn co-ordination in 2DJR, and dimerisation in 3SWP, and corresponding putative residues that may be involved in ZnF-BED1. Residues are labelled according to ZnF-BED1.

therefore refer to *ZnF-BED1* as *Rph7* for the remainder of the manuscript.

Sequence comparison across the five expressed genes at the locus between Cebada Capa and a further five barley cultivars predicted to carry *Rph7* (Ellinor, Toddy, Galaxy, Dictator 2 and La Estanzuela), confirmed they all shared the same haplotype. Further haplotypic comparisons between Cebada Capa and the 20 sequenced accessions comprising the barley pangenome[32] identified three distinct haplotypic groups (HI-HIII). Each of the haplotypic groups varying in gene content, corroborating with the previous haplotypic divergence reported by Scherrer et al.[29]. HI was similar structurally to Cebada Capa, HII contained mostly truncated homologues of only *Rph7* and *UnkP-1*, whereas HIII accessions lacked all pathogen-induced genes at the *Rph7* locus (Supplementary Fig. 1).

Accessions Barke, Hockett, RGT and HOR3365 carry full-length homologous DNA sequences where numerous SNP and indel polymorphisms are restricted to the first exon of *Rph7* at the N-terminus (Supplementary Fig. 2). However, the sequences likely transcribe into truncated short proteins. Barke, Hockett and RGT carry a SNP in the splice site at the end of exon 1 at the intron-exon boundary, which likely causes alternated splicing resulting in a truncated protein of 194 amino acid (AA) residues in length. The predicted protein sequence of Rph7 from HOR3365 is only 85 AAs due to a frameshift caused by a 1 bp insertion compared to the *Rph7* gene sequence from Cebada Capa. These comparisons suggest that resistance is likely due to either a PAV of the *Rph7* gene or the splice site mutation that leads to a truncated protein in those accessions that carry a homologue (Supplementary Fig. 2).

## Rph7 protein structural analysis suggests NAC transcription factor functionality

To further investigate a potential molecular function of *Rph7* we sought to gain insights from protein structure. With advances in recently developed deep-learning structure prediction tools, we predicted the Rph7 structure using AlphaFold2[33] (Supplementary Fig. 3a). The predicted Rph7 structure predominantly consists of a central β-sheet surrounded by α-helices, though we observed that the prediction confidence (pLDDT) varied across the protein with only some regions predicted with high confidence (pLDDT >70, residues 60–152 and 222–271) (Fig. 2b, c). To limit potential bias in our model we also predicted Rph7 in the absence of template structures (Supplementary Fig. 3b) and found that the predictions only varied at the regions predicted with low confidence (RMSD following superimposition across the structures was 1.5 Å, Supplementary Fig. 3c). We then performed a structural search against the protein databank (PDB) using the Dali server[34]. Dali reports structural similarity by Z-score, where significant similarities are indicated by Z-scores >2. Two of the top five unique protein structures (Supplementary Table 5) that have structural similarity to Rph7 were the N-terminal DNA-binding domain of the NAC (NO APICAL MERISTEM (NAM), ATAF1–2 and CUP-SHAPED COTYLEDON (CUC2)) proteins, ANAC019 (the top structural match) from *Arabidopsis* and the stress-responsive NAC1 from rice. Though structural superimposition between Rph7 and the DNA-binding domain (NAC domain) of ANAC019 showed the similarity is limited to the N-terminus of Rph7 (Fig. 2d). To confirm our previous observation that the C-terminus is a putative zinc-finger BED domain, we did a structural search with the C-terminal region alone (residues 220–302). Unsurprisingly we found the top structural hits were C2H2-type ZnF domains, and it appears that a zinc co-ordination motif (C3H) is present in Rph7 (Fig. 2c). Taken together, Rph7 contains an N-terminal NAC domain and C-terminal zinc-finger BED domain separated by a long-disordered loop. Despite the recent identification of zinc-finger BED domains located at the N terminal of recently cloned rust resistance genes encoding NLRs from both wheat (*Yr5* and *Yr7*) and barley (*Rph15*), to our knowledge a resistance protein of this structure has not been reported in the Triticeae[1].

In plants, NAC proteins are a large family of transcription factors, and typically consist of a conserved ~150 amino acid N-terminal NAC domain that is capable of binding DNA and facilitates dimerisation, and a diverse C-terminal domain that typically functions as a transcription regulatory domain[35]. In *Arabidopsis*, ANAC019 has a largely positively charged surface patch due to a cluster of arginine and predominantly lysine residues (Supplementary Fig. 4c) that are responsible for interacting with the backbone phosphates of the DNA molecule. Of particular importance in ANAC019 is a single, highly conserved Arg residue (Arg88) shown via mutational analysis to be required for DNA binding[36,37]. Similarly, Rph7 shows a positive surface suggesting that like ANAC019 it may be capable of binding DNA (Supplementary Fig. 4a, b), though Rph7 lacks the conserved Arg[35]. Dimerisation in ANAC019 is largely mediated by two prominent salt bridges formed by conserved R19 and E26 resides, which are conserved in Rph7 sequence and additionally localise in a similar region on the predicted Rph7 model (Fig. 2c). We predicted a putative dimeric Rph7 using AlphaFold2-multimer[38], though the interface pTM score (ipTM) was <0.2 indicating low model confidence that did not represent the experimentally determined dimeric ANAC019 structures. Despite this, given the conservation in residues it is tempting to speculate that the NAC domain of Rph7 dimerises and binds to DNA in a similar manner as ANAC019 though this remains to be experimentally determined.

To further understand the implications of the previously identified chemically induced mutants we mapped each of the sequence confirmed Rph7 mutants (G72E, T90I, R188*, and D209N) to the predicted structure (Fig. 2b). All four Rph7 mutants were surface exposed in the model and localise predominantly to flexible regions

within the protein, or to regions with a poor confidence prediction (Fig. 2b). The introduction of a premature stop codon at R188 would result in the production only of the NAC domain with an extended C-terminus, suggesting that it would retain the capacity to bind to DNA and presumably oligomerise but lacks the C-terminal ZnF-BED/regulatory domain. D209 occurs within the poorly predicted region of the AlphaFold2 model between the NAC and BED domain. Two of the mutants G72E and T90I localise to the opposite side of the protein, away from the putative DNA-binding surface suggesting these mutants would likely not impact this function directly. Though G72 localises close to R20 and E25 in the structure, it perhaps could be involved in mediating dimerisation, should Rph7 function similarly to ANAC019.

## *Rph7* expression regulates basal defence and induces partial cell death

Plant basal defence involves pathogenesis-related (*PR*) gene expression mediated by pathogen-induced transcription factors (TFs). In a separate experiment using qRT-PCR we re-confirmed that *Rph7* was weakly expressed on day one and more strongly expressed on day six in response to *Rph7* avirulent (220 P+ and 5457 P+) but not virulent (5553 P+) *P. hordei* pathotypes (Supplementary Fig. 5). This suggests that *Rph7* likely plays a critical role in signal transduction whereby virulent *P. hordei* pathotypes may lack effector(s) to induce the *Rph7* expression, resulting in susceptibility to leaf rust. In *Arabidopsis* stress-responsive NAC TFs respond to phytohormones, such as salicylic and jasmonic acid at the infection site, resulting in the expression of *PR* genes to induce the production of antifungal proteins and enzymes. We also used the RNA-Seq data generated in this study to perform a differential gene expression (DEG) analysis comparing the transcriptomes of Bowman and the BW758 near isogenic line (NIL) from the two timepoints. Unsurprisingly DEGs were substantially higher at day 6 in BW758 mirroring the expression of *ZnF-BED1* in *Rph7*-carrying lines (Fig. 3a). In parallel, Gene Ontology enrichment analysis revealed multiple biological processes related to the activation of basal plant defence (Supplementary Fig. 6). Furthermore, DEGs were identified by comparing BW758 and Bowman six days after infection. Four gene classes previously associated with host defence in response to fungal attack including: Jasmonate-related (*JR*), pathogenesis related (*PR*), *WRKY* transcription factors and *NLR* genes (Supplementary Fig. 7). This data suggests that NAC TFs like Rph7 play an important role in regulating or modulating cellular plant defence responses.

The recently cloned race-specific leaf rust resistance gene *Rph3*, like *Rph7*, is also specifically induced by avirulent races of *P. hordei* and was reported to encode a putative executor protein due to similarity of expression profile with other executor resistance proteins from rice[16]. We performed transient heterologous expression of *Rph7* (YFP-tagged) driven by CaMV 35S promoter in *N. benthamiana* and found that it alone could induce a partial cell death response relative to the Sr27/AvrSr27 control (Fig. 4a, b). Cell death was also measured in Golden Promise barley protoplasts which corroborated the *N. benthamiana* results where *Rph7* expression induced partial cell death relative to the positive Sr50/AvrSr50 control based on a significant increase in luciferase activity (Fig. 4c). Based on our AlphaFold structure prediction, Rph7 lacks the characteristic transmembrane features of the characterised executor proteins cloned to date (including Rph3)[16]. Nevertheless, the expression profile data we present suggests that the immune response in Cebada Capa mediated by *Rph7* is pathogen induced, thus implying that avirulent *P. hordei* pathotypes encode the same secreted effector that may function as a transcriptional activator-like effector (TALE)[16].

Interestingly, despite the characteristic near immune response elicited by most *Rph7*-avirulent *P. hordei* pathotypes, surveillance studies in Australia identified a group of pathotypes that elicited a

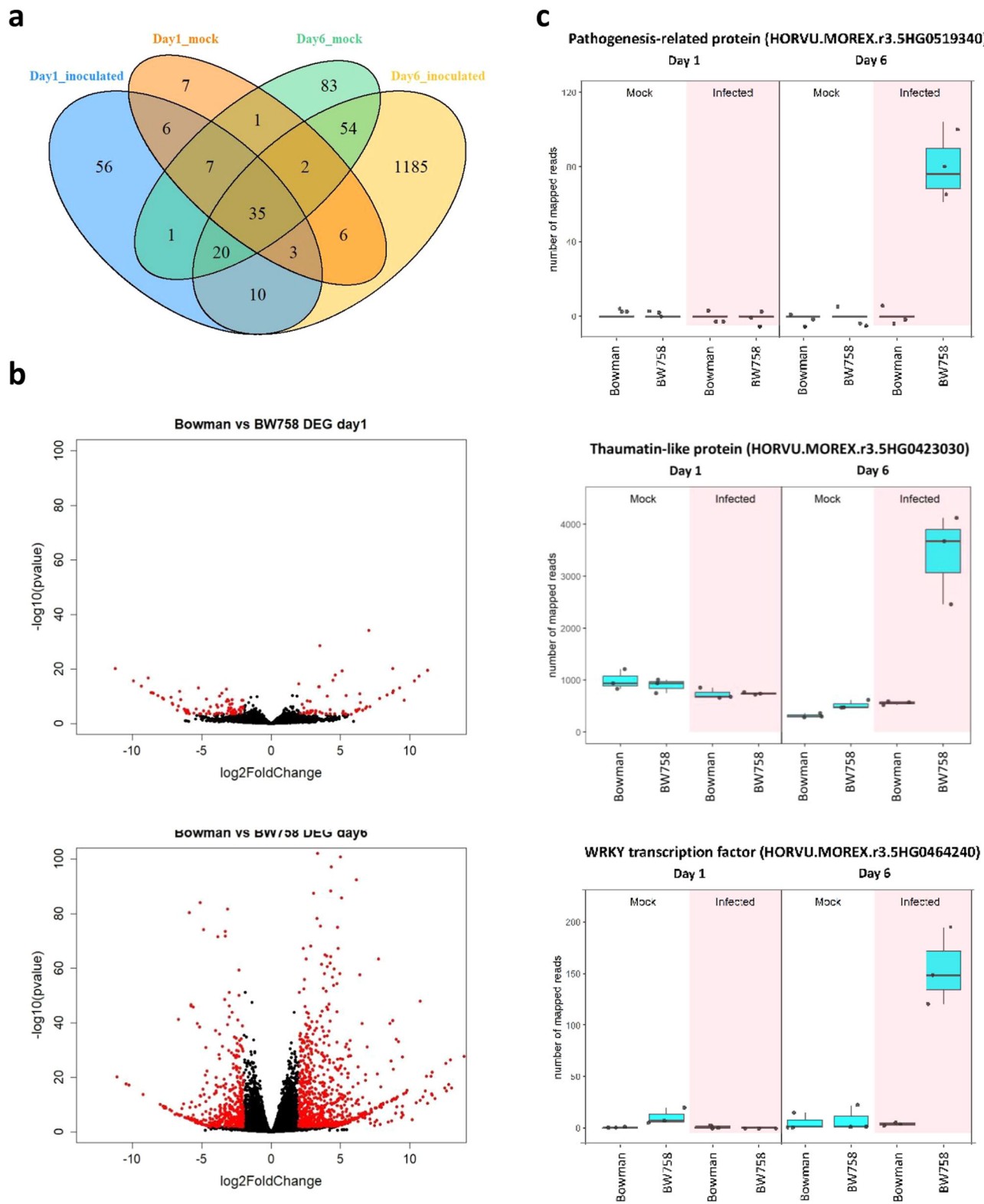

**Fig. 3 | Differential Expressed Genes (DEGs) detected between Bowman and BW758 (Bowman+ *Rph7*). a** Overview of the number of DEGs detected for treated (infected with *Rph7*-avirulent *Puccinia hordei* pathotype 5457 P+) and untreated (mock oil) at day 1 and day 6 after inoculation. The comparison identified 56 DEGs at day 1 and 1185 DEGs at day 6 associated with the rust inoculation treatment. **b** Volcano plots showing detected DEGs at day 1 (left) and day 6 (right). DEGs with Log2 Fold change > 2 or <-2 are highlighted in red. **c** Examples of detected key disease response marker genes showing upregulation at day 6 of infection that are predicted to encode pathogenesis-related protein, thaumatin-like protein and a

WRKY transcription factor. Samples labelled with light blue background were treated as mock control, while purple background labelled shows the expression patterns are from barley leaves infected with *Rph7*-avirulent *Puccinia hordei* pathotype 5457 P+. The data presented captures biological replication (*n* = 3) where the box encompasses two middle quartiles, with central line showing mean. Whiskers extend to the furthest data point within 1.5 times the interquartile range. A full list of DEGs involved in basal plant defence were provided in Supplementary Fig. 6. Source data are provided as a Source Data file.

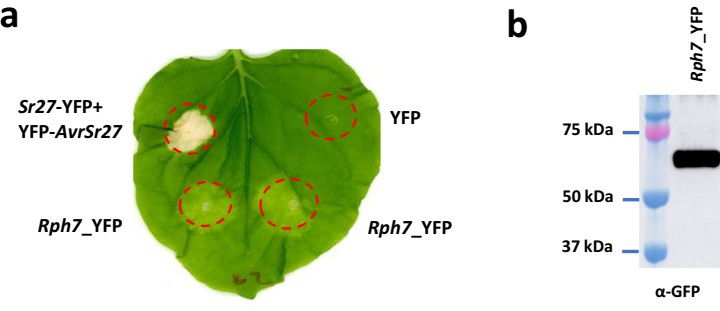

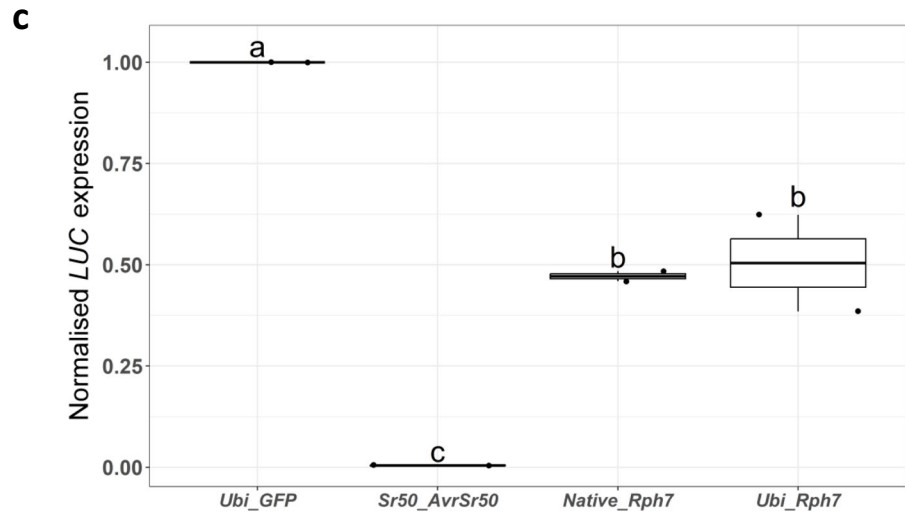

**Fig. 4 | Rph7 induces a cell death response when transiently expressed in _Nicotiana benthamiana_ and Golden Promise barley protoplasts. a** Transient expression of _Rph7_ (YFP-tagged driven by 35S CaMV promoter) induced partial cell death. The experiment was repeated three times and included infiltration of two leaves of three to four plants with similar results. The cloned stem rust resistance NLR protein Sr27 in the presence of its corresponding avirulence protein AvrSr27 was used as a positive control for cell death, and YFP alone as a negative control. **b** Western blot analysis of _Rph7_YFP expression in _N. benthamiana_. **c** Relative luciferase activity in barley (_Hordeum vulgare_ L. cv. Golden Promise) protoplasts from transgenic lines were co-transformed using the same _Rph7_ constructs shown in **a** with a luciferase reporter. Values are relative to luciferase activity of cells co-transformed with GFP and the luciferase reporter, which was set to 1. The cloned stem rust resistance NLR protein Sr50 in the presence of its corresponding avirulence protein AvrSr50 were used as a positive control for cell death. The data presented captures biological replication ($n = 2$) where the box encompasses two middle quartiles, with central line showing mean. Whiskers extend to the furthest data point within 1.5 times the interquartile range. Samples marked by identical letters in the plots do not differ significantly ($P < 0.01$) using the single-tailed Tukey test. Source data are provided as a Source Data file.

necrotic intermediate response, and more recently, a single isolate that was fully virulent on _Rph7_. This initially led to the hypotheses that either two or more genes may confer _Rph7_-mediated resistance or alternatively that distinct _P. hordei_ isolates eliciting a differential _Rph7_ response were either homozygous or heterozygous for the avirulence gene matching _Rph7_. Our data rules out the two resistance gene hypothesis, however, genetic analyses based on crosses of _P. hordei_ pathotypes (immune × intermediate _Rph7_ response) will be required to test the heterozygous vs homozygous avirulence hypothesis. Once resolved further studies should involve the cloning and sequence comparison of the corresponding effector/s for _Rph7_ between _P. hordei_ pathotype groups eliciting immunity and hypersensitive cell death responses (Fig. 1d). This in parallel with further mechanistic studies on the downstream targets and regulation of the _Rph7_ resistance will best equip biotechnologists to efficiently engineer crop plants for durable disease control.

## Methods
### Plant materials
A high resolution F$_2$ fine mapping population ($n = 2200$ gametes) was developed by intercrossing Cebada Capa (_Rph7_) with WABAR 2722

(_rph7_). The NIL BW758 (Bowman+_Rph7_) was used for mutagenesis as well as differential gene expression experiments in comparison to the wild-type cv Bowman.

### Pathogens and phenotypic analysis
Details of the four _P. hordei_ pathotypes used in this study, including pathogenicity on different resistance genes, are listed in Supplementary Table 3. Pathotypes were designated according to the octal notation proposed by Gilmour[36] and Park and Karakousis[37]. _Rph7_- avirulent pathotype 5457 P+ was used to phenotype the mapping population for recombinants, screening mutants and RNA-Seq expression analysis. _Rph7_- avirulent (200 P+, 276 P+ and 5477 P−) and virulent (5553 P+) pathotypes were used to validate resistance gene function and specificity in the T$_1$ generation of Golden Promise + _Rph7_ transgenic families. These pathotypes were isolated and increased[16] before being stored in liquid nitrogen at the Plant Breeding Institute, the University of Sydney, Australia. Rust testing of barley seedlings in the greenhouse with _P. hordei_ pathotypes listed above was performed[16] and plants were phenotyped 10 days post inoculation using the "0"–"4" infection type (IT) scale[3].

## Mutagenesis

A mutant population was developed by chemically treating 1000 BW758 seeds using sodium azide[38] with some modifications. Seeds were wrapped in cheesecloth, immersed in water at 4 °C overnight and then transferred into 2-litres of water to aerate with pressurised air for 8 h followed by draining. Seeds were treated on a shaker with 1 mM sodium azide dissolved in 0.1 M sodium citrate buffer (pH 3.0) on a shaker for 2 h, washed under running water for at least 2 h, and then dried in a fume hood overnight. Seeds were space planted in the field directly and harvested as described by Chen et al. [39]. More than 4500 $M_2$ generation spikes were phenotyped by rust testing with *P. hordei* pathotype 5457 P+ as clumps at the seedling stage for segregating *rph7* knockouts[13,39]. For MutChromSeq analysis, non-amplified DNA of 3H chromosome of BW758 and seven susceptible mutants were shotgun sequenced after flow-cytometric sorting using the NovoSeq[13,39]. The wild-type reference sequence was assembled using Meraculous and candidate gene identification was performed according to Sanchez-Martin et al.[31]. Briefly, the reads of wild type and mutants were aligned to de novo chromosome reference sequence of BW758 using bwa v0.7.17 and SNPs were called using SAMtools v1.9. MutChromSeq pipeline (https://github.com/steuernb/MutChromSeq) was used to identify contigs with mutation in multiple mutant lines. All contigs with mutations in three and more independent mutants were investigated manually in IGV and candidate genes were revealed and characterised by homology (blastx) search against barley proteins.

## RNA-Seq data preparation

Cebada Capa, Bowman + *Rph7* (BW758) and Bowman seedlings were grown in trays[16] and then inoculated using either oil (mock control) and oil mixed with 30 mg of *P. hordei* urediniospores (pt. 5457 P+). Leaf tissue (three technical replicates for each of the three biological replicates per genotype) was harvested 24 hours (day 1) and 6 days post inoculation (dpi) and snap frozen in liquid N and stored at −80 °C (Supplementary Table 6). RNA was isolated using the Maxwell robot (Promega). RNA quality and quantity was assessed by RNA gel electrophoresis and the Nanodrop spectrophotometer. Illumina paired end 150 bp sequencing was performed by Novogene yielding between 21 and 37 million raw reads per sample. Adaptor and quality trimming of raw reads was performed using fastqc version 0.11.9[40] and trimmomatic 0.39[41], respectively.

## Candidate gene identification

Genotypic screening for recombinants was performed using PCR-based markers or KASP markers (Fig. 1a, Supplementary Table 7). Critical recombinant $F_3$ families were progeny-tested both phenotypically and genotypically (12 plants per family) (Supplementary Table 1). The sequenced section between the conserved genes *HvGAD1* and *HvHGA1* of the Morex_V2[42] reference sequence was replaced with sequenced BACs AY642925 and AY642926[29] to avoid background mapping of similar reads within the transcriptome sequences. Cleaned RNA-Seq reads were mapped with Tophat 2.1.1[43] and Bowtie2 2.4.4[44] using default settings. Read depth for each base pair was calculated with the depth function of SAMtools version 1.12[45]. Mean read coverage of biological replicates were visualised with R bar plot to identify expressed sequence regions.

## In silico expression analysis

De novo transcriptome assembly of Cebada Capa was performed with trinity version 2.13.2 (https://github.com/trinityrnaseq/trinityrnaseq, Grabherr et al.[46]). We selected one replicate from day 1 and day 6 of infected and uninfected Cebada Capa to generate a Cebada Capa specific reference (B4, B27, C4 and C28, Supplementary Table 6). These were pooled before assembly. Gene and isoform expression levels were estimated with RSEM version 1.3.3[47] using trinity mode (https://github.com/deweylab/RSEM/releases). Contigs of the trinity assembly have been matched with candidate genes by BLASTn and read counts visualised by R boxplot.

## Differential gene expression analysis

Purified RNA-Seq reads have been mapped against Morex_V3[48] with HiSat2 version 2.2.1[49]. BAM alignment files have been sorted with SAMtools 1.12 (https://github.com/samtools/samtools/, Danecek et al.[45]). Reads mapped to transcripts using gff3 file of Morex V3 annotation (July 2020) by htseq-count[50]. Differential gene expression analysis has been performed with DEseq2 R package[51]. Genes expressed with a log2 fold change >2 or <−2 have been subjected to gene ontology enrichment using the Triticeae-Gene Tribe database (http://wheat.cau.edu.cn/TGT/)[15].

## Candidate gene cloning, vector construction and barley transformation

Cloning and *Agrobacterium*-mediated transformation of a 3625 bp genomic fragment was performed as described in Chen et al.[39]. Briefly, primers were designed to amplify the *ZnF-BED1* gene from Cebada Capa including the sequence of the native promoter and terminator that are 1.5 kb upstream of the ATG and 1 kb downstream of the STOP codon respectively. The PCR fragment was cloned into TOPO cloning vector XL-2 and sequence was confirmed by Sanger sequencing. The PCR fragment was then sub-cloned into pMB-vec8 vector using the *Not* I restriction enzyme site. The confirmed positive plasmid was transformed into *Agrobacterium tumefaciens* strain AgL1.

## Structural modelling and structural comparisons of ZnF-BED1

Five structural models of ZnF-BED1 were generated using Google DeepMind's AlphaFold2[33]. Full databases were used for multiple sequence alignment (MSA) construction. All templates downloaded on July 20, 2021, were allowed for structural modelling in the first instance and excluded in the second prediction to reduce potential bias. We selected the best model (ranked_0.pdb) for downstream protein visualisation using open-source Pymol or UCSF *ChimeraX*[52]. Structural comparisons to known structures in the protein databank were carried out using the Dali webserver[34].

## Transient expression of *Rph7* in *Nicotiana benthamiana* leaves and barley protoplasts

*Rph7* driven by maize polyubiquitin promoter and terminator from the *Agrobacterium* tumour morphology 1 gene (*Ubi_Rph7*): *Rph7* fragment was amplified using the primer pair *Rph7_attB1*/*Rph7_attB2* and genomic DNA from *Rph7*-carrying wild-type barley cultivar Cebada Capa. The fragment was purified from the gel and BP reaction was set up by adding the fragment, pDON207 and the BP Clonase (Thermofisher). The plasmid DNA (pDON207_*Rph7*) from the positive colonies was extracted and verified by sequencing. LR reaction was set up by adding pDON207_*Rph7* and *Bar_Ubi* gateway vector (provided by Ming Luo) and LR clonase. The positive clones were verified by gene-specific PCR and sequencing. *Rph7* cDNA driven by 35S CaMV promoter and a NOS terminator (*Rph7_YFP*) was synthesised from the RNA sample 6 days after avirulent pathogen induced, which is the same from the expression analysis. cDNA was used as template to amplify *Rph7* using primer pairs *Rph7*_ATG attB1 and *Rph7*_END attB2. The fragment was purified from the gel and BP reacted to pDON207. The positive clones were verified by sequencing and then LR reacted into pAM-YFP vector (*Rph7_YFP*) which is fusion with YFP at C-terminus.

The infiltration of *N. benthamiana* leaves was performed following the protocol described by Ortiz et al.[53]. In brief, the *N. benthamiana* plants were grown at 25 °C under 16 h light and 8 h darkness period for 4 weeks. Plasmids *Native_Rph7* and *Ubi_Rph7* were transformed into *Agrobactium tumefacians* strain AgL1. *A. tumefacians* cultures were incubated for 24 h, centrifuged at 2500 × *g* and the pellet was resuspended in MMA buffer (10 mM MES, pH 5.6, 10 mM $MgCl_2$, 150 µM

Acetosyringone) and incubated at room temperature for 2 h. Prior to infiltration, the OD for each of the treatment constructs was adjusted to 1.0. The combination of *Sr27* and *AvrSr27* was used as a positive control and YFP as a negative control[54]. All images were taken five days after infiltration. The samples (*Rph7_YFP*) were harvested 24 h after infiltration for the western blot assay using a 1:2000 dilution of anti-GFP antibody (mouse IgG clones 7.1 and 13.1).

Transient expression in protoplasts of cv. Golden Promise barley was following the protocol described by Saur et al.[55]. Briefly, protoplasts were isolated from barley cv. Golden Promise grown under low light for eight days. Co-expression of *Sr50* with *AvrSr50* and the reporter gene luciferase was used as a positive control for cell death, while co-expression of *Ubi-GFP* and *Ubi-luciferase* as a negative control[56]. *Native-Rph7*, *Ubi-Rph7*, and *Rph7-YFP* were transiently co-expressed with *Ubi-luciferase* in barley protoplasts, and the luciferase activity was measured 20 h later. The data were from two biological replicates with three technical repeats and were normalised to relative expression of luciferase to the negative control (*Ubi-luciferase* and *Ubi_GFP*) which is set at 1.00. A Tukey test was performed to determine how statistically significant the means of each treatment were from each other.

### Reporting summary

Further information on research design is available in the Nature Portfolio Reporting Summary linked to this article.

## Data availability

The *Rph7* mRNA has been added to NCBI under accession number OR083044. The raw reads of the RNA-Seq experiment have been deposited to NCBI SRA archive under BioProject PRJNA924522. The raw reads of Mutchromseq were submitted to NCBI SRA archive under BioProject ID PRJNA906712 [https://www.ncbi.nlm.nih.gov/bioproject/?term=rph7]. Source data are provided with this paper.

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

## Acknowledgements

The authors thank Beat Keller and Gerhard Herren for scientific discussion about details of the previous published work. We thank B. Clark, L. Ma, M. Williams, and S. Hoxha for technical supports with plant growth, molecular experiments, mutagenesis, and pathogen spore increases. We thank the GRDC for supporting the research, P.M.D., C.C., M.J., D.F. were funded under grant UOS1507-005RMX - US00074 awarded to R.F.P. and E.L. P.M.D was also supported by La Trobe University and the Alexander von Humboldt Foundation. M.A.O. was supported by the CSIRO Research Postdoctoral Fellowship. We thank Zdeňka Dubská, Romana Šperková and Jitka Weiserová for preparation of chromosome samples for flow cytometry and Petr Cápal and István Molnár for chromosome flow sorting. The authors acknowledge the use of CSIRO High Performance Computing Facility and expertise provided by the CSIRO IMT Scientific Computing, and Michael Kuiper (Data61) for providing AlphaFold support.

## Author contributions

R.F.P., E.L. and P.M.D. conceived the project. R.F.P. provided all rust isolates, information on pathogenicities and barley genetic mapping populations. D.S. and P.M.D. phenotyped the recombinants. C.C., P.M.D. and M.J. fine mapped the *Rph7* locus. C.C., M.J and P.M.D annotated the final locus of *Rph7*. P.Z. and P.M.D. created mutant materials, P.M.D. screened and confirmed the knockout mutants. J.D., J.B. and K.H. flow sorted and sequenced wild type and mutant chromosomes and performed MutChromSeq analysis. M.J. designed and performed the haplotype and pan-genome analysis. C.C. performed the gene amplification, vector construction and cloning of *Rph7* for the complementation experiment and expression analysis using qRT-PCR. S.P., C.C., M.J. and P.M.D. designed the RNA-Seq experiment, D.F, and P.M.D experimented, M.J. and C.C. analysed the data. P.M.D., C.C. and D.F. performed all Sanger Sequencing confirmation and follow up alignment analysis for mutants on candidate genes at the *Rph7* locus. D.B. created the transgenic material, and D.S. multi-pathotype tested transgenic progeny. C.C., J.C. and A.W. designed and performed transient expression analysis. M.A.O. performed all protein structure prediction analysis. P.M.D., E.L. and R.F.P. supervised the project. P.M.D. wrote the manuscript with contributions from C.C., M.J and M.A.O. All authors reviewed and commented on the manuscript.

## Competing interests

The authors declare no competing interests.
