## [Peer Review File · Nature Communications]

A pathogen-induced putative NAC transcription factor mediates leaf rust resistance in barleyReviewers' Comments:

Reviewer #1:

Remarks to the Author:

This manuscript describes the molecular identification and characterization of the Rph7 resistance gene of barley against the biotrophic fungal pathogen leaf rust. Based on an earlier mapping study, additional genetic mapping combined with RNAseq analysis allowed the authors to identify novel candidate genes. Using mutagenesis and transformation as validation tools, Rph7 was successfully identified. It is located in a genetically distinct haplotype that is characteristic for lines containing this resistance gene. Rph7 encodes a putative NAC transcription factor and belongs to a gene class that has not previously been implicated in race-specific disease resistance. Interestingly, Rph7 is only induced by avirulent pathogen isolates, but not by virulent leaf rust races.

The work adds to our increasing knowledge on molecular diversity of race-specific resistance mechanisms in the Triticeae crops barley and wheat. The work is technically well done and provides convincing evidence about the identity of the Rph7 resistance gene. The authors have recently published the Rph3 gene which is also specifically induced by avirulent races of leaf rust, and therefore could act an executor gene. It would be interesting to know if the authors have any data which might indicate executor gene function of Rph7. For example, it would be interesting to test if transient expression of Rph7 in *N. benthamiana* results in cell death symptoms. A similar experiment could be performed in barley protoplasts. It also remains unclear from the manuscript how the authors conclude on the involvement of Rph7 in basal resistance. This would certainly not explain a strong race-specific resistance.

Additional points:

For some aspect, the text does not provide sufficient details on the analysis.

L114 ff: MutChromSeq on how many mutants? How many contigs contained mutations? How were the MutChromSeq data used to fill the gap between the two BAC clones?

L145ff: please explain in detail the splice site variant and the variable partial variants of Rph7.

L197: What is "not virulent" in contrast to "avirulent"? The text needs clarification

Figure 1D: Instead of labeling a-f, it would help the reader to directly indicate barley genotype, and isolate used.

Figure 2: Much of the information in Supp Figure 2a could be added here: For example, the mutations could be added in A. Similarly, the rest of supp figure 2 could be part of figure 2

It would be helpful to show ANAC019 structure below B in Figure 2, in addition to the overlay in C.

There is a major problem in Figure 3C. How can it be that the same gene, 6 days after infection, in the same genotype BW758 shows two completely different expression levels?

Extended data figure 2: Is H1 = I in the figure? Please use uniform nomenclature for haplotypes.

Figure Legend extended data figure 6: There is no B in the legend

Supplementary table 1 is not sufficiently explained (H, R)

Reviewer #2:

Remarks to the Author:

I was asked to be a reviewer as an expert in structural biology, thus I focus on this particular aspect of the paper only.

Generally, I find the work interesting, as it regards the identification of a transcription factor implicated in plant pathogen resistance, which is a topic of great importance.

I also commend the authors for trying to bridge the gap between their genetic analysis and molecular structure through use of the powerful AlphaFold structure prediction methods.

I have a few technical issues and questions:

1) As I understand from methods description, AF2 was allowed to use templates, were any of the DALI matches used as templates? That would still make the result significant, but obviously if the same structures are used as templates there is a certain bias. Did AF2 give a similar prediction without templates?

2) I personally severely doubt that the low confidence regions in the AF2 model resemble anything like what shown. I appreciate that the authors want to show the results, but this could be in extended data. A main figure could just include higher confidence regions (this would help not mislead less expert readers). It would be helpful to have the sequence color coded according to model reliability too.

3) Page 6: the average score is very uninformative. I think it would be most informative to be told which sequence regions give a very highly reliable model, and forget about the rest.

4) Supp table 5, the first 3 hits are different structures of the same protein. The legend is thus not appropriate (unique proteins) and the referring text is also not correct.

5) Line 160 not appropriate to refer to the origin of the NAC name (acronym of initials of NAM, ATAF1,2, and CUC2) at such a late stage and without explanation - it should be defined earlier. Particularly because in the context of this text, it sounds like NAM, ATAF1,2 and CUC2 are the actual identities of the DALI hits, this is not the case.

6) Extended data figure 4 lacks appropriate labelling – and I suggest to limit to region modeled reliably by AF2.

Does the model fit with known DNA binding region in ANAC019

Would dimerization occur? One could try and model the dimer or see if dimerization interface is reasonable

Why do the authors assume a flexible linker? Just based on poor AF2 score? It needs to be explained

7) Ideally hypotheses about the molecular structure and function should be confirmed by expressing the protein in vitro or making in vivo mutants based on the structure. I do accept though that this is outside the main scope of the work.

I definitely agree with the main conclusions that this is a NAC transcription factor, possibly with a Zn finger attached. However all functional properties of the two domains would need to be verified. If production of purified protein is impossible, perhaps in vivo mutagenesis guided by the structure could be attempted, but I do not know if this is realistic.

Reviewer #3:

Remarks to the Author:

The study identified a barley leaf rust resistance gene Rph7. Briefly, fine mapping was performed to an interval with five genes. RNA-Seq was generated for resistance and susceptible lines with and without the treatment of multiple pathotypes. Mutagenesis and stable transformation were used to confirm the gene that encodes a protein with a putative NAC domain and a zinc finger BED domain.

The study provided solid evidence to support the cloning of the resistance gene. No major concern about this study although I feel that the writing of the manuscript could be improved. For example, when describing expression regulation, it would be helpful to be clear on what comparison was referred to. Some minor comments are listed below.

1. It was mentioned that Rph7 is a semi-dominant allele. However, transgenic lines seem to become a dominant allele (P128: 3 resistance : 1 susceptibility). Additional comment about the phenotypic

difference between the native allele and the transgenic allele will be appreciated.

2. The Abstract is poorly written.

1) The following sentence includes rich information but is unclear: "We identified three up-regulated and pathogen-induced genes with presence/absence variation (PAV) at this locus that were only expressed in response to Rph7 avirulent pathotypes of *P. hordei*."

2) "Progeny from four independent transgenic lines segregated for the expected avirulent Rph7 infection type in response to several avirulent *P. hordei* pathotypes, however, all plants were susceptible to a single virulent pathotype confirming the specificity." could be reworded to clarify the statement.

3) It would be helpful to add a note about ANAC019 to indicate what it is.

3. P107: progeny-tested eight susceptible knockout M4 families

It would be informative to describe the EMS screening result. Some description in the Methods (e.g., Ten M2 families were identified as putative Rph7 knockout mutants.) could be moved to here.

4. P115: Description of MutChromSeq can help the audience understand the method.

5. P141: "...three distinct haplotypic groups: H1-H3. each varying in gene content corroborating previous haplotypic data reported by Scherrer, et al. 29." This sentence needs a correction.

6. P209: Expression of four groups of genes (JR, PR, WRKY, and NLR) was focused. But no description about WRKY or NLR in the main text. It would be interesting to understand how the responses of NLR.

7. P296 "Genes expressed with a log2 fold change $>+/-2$ " is not correct. If log2 is a negative value, the criterium should be less than -2.

REVIEWER COMMENTS

Reviewer #1 (Remarks to the Author):

This manuscript describes the molecular identification and characterization of the Rph7 resistance gene of barley against the biotrophic fungal pathogen leaf rust. Based on an earlier mapping study, additional genetic mapping combined with RNAseq analysis allowed the authors to identify novel candidate genes. Using mutagenesis and transformation as validation tools, Rph7 was successfully identified. It is located in a genetically distinct haplotype that is characteristic for lines containing this resistance gene. Rph7 encodes a putative NAC transcription factor and belongs to a gene class that has not previously been implicated in race-specific disease resistance. Interestingly, Rph7 is only induced by avirulent pathogen isolates, but not by virulent leaf rust races.

The work adds to our increasing knowledge on molecular diversity of race-specific resistance mechanisms in the Triticeae crops barley and wheat. The work is technically well done and provides convincing evidence about the identity of the Rph7 resistance gene. The authors have recently published the Rph3 gene which is also specifically induced by avirulent races of leaf rust, and therefore could act an executor gene. It would be interesting to know if the authors have any data which might indicate executor gene function of Rph7. For example, it would be interesting to test if transient expression of Rph7 in *N. benthamiana* results in cell death symptoms. A similar experiment could be performed in barley protoplasts. It also remains unclear from the manuscript how the authors conclude on the involvement of Rph7 in basal resistance. This would certainly not explain a strong race-specific resistance.

We would like to thank the reviewer for their positivity regarding the merits of the manuscript and suggestion for the follow-up experiments to test whether transient expression of Rph7 in tobacco and possibly barley protoplasts results in cell death. We performed both of these experiments since the first submission and have added the results to the manuscript including a new figure (Figure 4) in the main section of the manuscript and also supporting text both describing and interpreting the data.

“The recently cloned race-specific leaf rust resistance gene Rph3, like Rph7, is also specifically induced by avirulent races of P. hordei and was reported to encode a putative executor protein due to similarity of expression profile with other executor resistance proteins from rice¹⁶. We therefore performed transient heterologous expression of Rph7 (YFP-tagged) driven by CaMV 35S promoter in N. benthamiana and found that it alone could induced a partial cell death response (Fig. 4A, 4B). Cell death was also measured in Golden Promise barley protoplasts which corroborated with the N. benthamiana results where Rph7 expression induced partial cell death relative to the positive Sr50/AvrSr50 control based on a significant increase in luciferase activity (Fig. 4C). Based on our AlphaFold structure prediction, Rph7 lacks the characteristic transmembrane features of the characterised executor proteins cloned to date (including Rph3). Nevertheless, the expression profile data for Rph7 we present suggests that it only triggers and enhances the immune response in barley when challenged with P. hordei pathotypes that likely all carry the same avirulence gene encoding a secreted a transcriptional activator-like effector (TALE).”

We agree that based on the literature, enhanced basal resistance is not usually a characteristic or the main common feature of race-specific resistance mechanisms in plants. We further clarified the overall conclusive hypothesis in regard to *Rph7* mediated resistance and the resulting enhanced defence response in *Rph7* carrying lines. From our RNA-Seq data several pathogenesis related genes showed identical expression profiles to the *Rph7*

gene, i.e. several PR genes were only expressed at day 6 in *Rph7* carrying lines (Cebada Capa and near isogenic line BW758- Bowman + *Rph7*) after being challenged with the pathogen. Given *Rph7* encodes a putative NAC transcription factor based on our *in silico* data we hypothesise that it acts as a master regulator as part of a complex with another unknown target to enhance resistance and this either directly or indirectly enhances basal resistance. It should also be noted that virulence for the *Rph7* resistance is even rarer than for *Rph3* with only one very recent isolate identified in Australia.

Additional points:

For some aspect, the text does not provide sufficient details on the analysis.

L114 ff: MutChromSeq on how many mutants? How many contigs contained mutations?

We modified the text (please see below) to include the mutant number (seven) in the text.

“All contigs with mutations in three and more independent susceptible mutant lines were manually examined. There were 2677 such contigs representing about 3% of all assembled contigs. While the majority contain no genic sequence or unexpected mutations not induced by chemical treatment (sodium azide produces C>T or G>A conversions), a single contig contained mutations in gene-coding sequence in three independent lines.”

How were the MutChromSeq data used to fill the gap between the two BAC clones?

The MutChromSeq data was not used to fill the gap between the two BAC clones, rather we performed the analysis as support for our mutant sequencing and also to confirm or rule out the presence of additional candidates for *Rph7* on chromosome 3H. The authors have amended the text accordingly to clarify this in the manuscript.

“Due to the lack of a continuous sequence at the resistant haplotype, we opted to perform a MutChromSeq experiment to add a second layer of confidence not missing putative further candidates. MutChromSeq is an unbiased approach in complexity reduction used to rapidly isolate plant genes and regulatory DNA sequences that is not reliant on recombination-based genetic mapping¹³. We flow sorted and sequenced the 3H chromosomes of wild type and seven of the mutant lines and performed MutChromSeq analysis.”

L145ff: please explain in detail the splice site variant and the variable partial variants of *Rph7*.

The analysis and text in the manuscript is purely based on an *in silico* sequence alignment of the *Rph7* sequence from Cebada Capa with homologous sequences identified from fully sequenced chromosome-scale references of the barley pan-genome. The four accessions mentioned carry full length homologous sequences on a DNA level, however due to the identified splice-site variant at the end of exon 1, we speculate that this leads to a truncated non-functional *Rph7* protein and could partially explain the difference between resistant and susceptible accessions in accessions where a homologue exists. The authors, in light of the reviewer’s comments, have improved the clarity in the section and included details about resulting putative truncated proteins below.

*“Accessions Barke, Hockett, RGT and HOR3365 carry full-length homologous DNA sequences where numerous SNP and indel polymorphisms are restricted to the first exon of *Rph7* at the N-terminus (Extended Data Fig. 2). However, the sequences likely transcribe into truncated short proteins. Barke, Hockett and RGT carry a SNP in the splice site at the end of exon 1 at the intron-exon boundary, which likely causes alternated splicing resulting*

in a truncated protein of 194 amino acid (AA) residues in length. The predicted protein sequence of Rph7 from HOR3365 is only 85 AAs due to a frameshift caused by a 1bp insertion compared to the Rph7 sequence from Cebada Capa. These comparisons suggest that resistance is likely due to either a PAV of the Rph7 gene or the splice site mutation that leads to a truncated protein in those accessions that carry a homologue (Extended Data Fig. 3)."

L197: What is "not virulent" in contrast to "avirulent"? The text needs clarification

We clarified this text accordingly.

Figure 1D: Instead of labeling a-f, it would help the reader to directly indicate barley genotype, and isolate used.

We have made the suggested change.

Figure 2: Much of the information in Supp Figure 2a could be added here: For example, the mutations could be added in A. Similarly, the rest of supp figure 2 could be part of figure 2

We have adopted the suggested changes and merged Supplementary Figure 2 into the main Figure 2.

It would be helpful to show ANAC019 structure below B in Figure 2, in addition to the overlay in C.

We have chosen not to include the ANAC019 structure separately here. This structure was not determined in this work and is in a publicly available database as identified in the Figure caption, and Supporting Table. For reproducibility, we have included the AlphaFold2 predictions for Rph7 (both the full prediction and high confidence only model), such that reproducible analyses can be carried out by readers.

There is a major problem in Figure 3C. How can it be that the same gene, 6 days after infection, in the same genotype BW758 shows two completely different expression levels?

We apologise for the incomplete figure legend which did not allow the correct interpretation of the figure 3C. The change refers to the difference between mock treated (oil and no pathogen inoculum) and rust pathogen inoculated which is shaded as purple background vs grey background. The figure legend has been updated to include the missing information.

Extended data figure 2: Is H1 = I in the figure? Please use uniform nomenclature for haplotypes.

We have now adopted uniform nomenclature.

Figure Legend extended data figure 6: There is no B in the legend

We have addressed this typo.

Supplementary table 1 is not sufficiently explained (H, R)

We have made alterations to sufficiently explain the data in Supplementary Table 1.

Reviewer #2 (Remarks to the Author):

I was asked to be a reviewer as an expert in structural biology, thus I focus on this particular aspect of the paper only.

Generally, I find the work interesting, as it regards the identification of a transcription factor implicated in plant pathogen resistance, which is a topic of great importance.

I also commend the authors for trying to bridge the gap between their genetic analysis and molecular structure through use of the powerful AlphaFold structure prediction methods.

I have a few technical issues and questions:

1) As I understand from methods description, AF2 was allowed to use templates, were any of the DALI matches used as templates? That would still make the result significant, but obviously if the same structures are used as templates there is a certain bias. Did AF2 give a similar prediction without templates?

We thank the reviewer for their comment and agree there is a certain bias with the inclusion of templates. To address this, we have included a list of all the PDB templates accessed by AlphaFold2 during the structure prediction process as an additional supporting table (Sup. Table 5. In this table we have also highlighted (in bold) any proteins that correspond to the DALI server output. For completion we also present the predicted structure of Rph7 without templates Extended Data Fig. 3C), as well as a comparison between the template-based and no-template based prediction (Supp. Fig. 3D). We have also included the following wording in text “To limit potential bias in our model we also predicted Rph7 in the absence of template structures (Extended Data Fig. 3B), and found that the predictions only varied at the regions predicted with low confidence (RMSD across the structure 1.5 Å, Extended Data Fig. 3C).” We have also adjusted the wording in the methods to include this information “All templates downloaded on July 20, 2021, were allowed for structural modelling in the first instance and excluded in the second prediction to reduce potential bias.”

2) I personally severely doubt that the low confidence regions in the AF2 model resemble anything like what shown. I appreciate that the authors want to show the results, but this could be in extended data. A main figure could just include higher confidence regions (this would help not mislead less expert readers). It would be helpful to have the sequence color coded according to model reliability too.

We agree with the reviewer that the low confidence regions are unlikely to resemble the true Rph7 structure. We have altered Figure 2B such that the main figure only shows the regions with high confidence (using a confidence cut off of >70%) and present the full structure in the Extended Data Fig 3A. As suggested above, we have also included a sequence as part of Figure 2C with the colour coding according to the confidence score.

The caption has been updated to “(B) AlphaFold2 prediction of ZnF-BED1. Top: Amino acid sequence representation of the Rph7 coloured according to the per-residue confidence score (pLDDT). Bottom: Regions of high confidence (per-residue confidence score (pLDDT) >70) only are shown as a cartoon representation. For full structure please see Extended data Fig S3. The PDB file of the full structure and pLDDT >70 are available as additional supporting files.

We have also altered the wording here “To further investigate a potential molecular function of Rph7 we sought to gain insights from protein structure. With advances in new deep-learning structure prediction tools, we predicted the Rph7 structure using AlphaFold2 (Extended Data Fig. 3A). The predicted Rph7 structure predominantly consists of a central β -sheet surrounded by α -helices, though we observed that the prediction confidence (pLDDT) varied across the protein, though only some regions predicted with high confidence (pLDDT >70, residues 60-152 and 222-271) (Fig. 2C).”

3) Page 6: the average score is very uninformative. I think it would be most informative to be told which sequence regions give a very highly reliable model, and forget about the rest.

We thank the reviewer for their comment, and agree that the average score can be misleading. We have included the amino acid sequence colour coded by confidence (see comment above) and have altered the wording to remove mention of the average score, as well as described the regions of high confidence, as discussed in the point above.

4) Supp table 5, the first 3 hits are different structures of the same protein. The legend is thus not appropriate (unique proteins) and the referring text is also not correct.

Apologies, this should have read unique structures not unique proteins deposited in the PDB. We have updated this table to show unique proteins (not unique structures) as we believe this is more informative please see updated Table S5), this now accurately reflects the caption and in text reference.

5) Line 160 not appropriate to refer to the origin of the NAC name (acronym of initials of NAM, ATAF1,2, and CUC2) at such a late stage and without explanation - it should be defined earlier. Particularly because in the context of this text, it sounds like NAM, ATAF1,2 and CUC2 are the actual identities of the DALI hits, this is not the case.

We thank the reviewer for their comment. We introduce the NAC acronym and define it here as it is the first reference in the main text, and it is not applicable to define the acronym earlier. To improve the clarity here as suggested we have altered the sentence from “*Four of the top five unique proteins (Z-score > 6) were NAC (NAM, ATAF1,2, and CUC2) domain-containing proteins (Supplementary Table 5).*”

This now reads, in conjunction with suggested text alterations above: “*Two of the top five unique protein structures (Supplementary Table 5) that have structural similarity to Rph7 were the N-terminal DNA-binding domain of the NAC (NO APICAL MERISTEM (NAM), ATAF1–2 and CUP-SHAPED COTYLEDON (CUC2)) proteins, ANAC019 (the top structural match) from Arabidopsis and the stress-responsive NAC1 from rice. Though structural superimposition between Rph7 and the DNA-binding domain (NAC domain) of ANAC019 showed the similarity is limited to the N-terminus of Rph7 (Fig. 2D).*”

6) Extended data figure 4 lacks appropriate labelling – and I suggest to limit to region modeled reliably by AF2.

We apologise for the lack of labelling this has been amended. We only now show the full structure (Extended Data Fig. 3A) as well as the regions of high confidence (pLDDT >70) (Extended Data Fig. 3B), as well as the NAC domain structure of ANAC019 (top Dali match to Rph7) (Extended Data Fig. 4C).

Does the model fit with known DNA binding region in ANAC019?

In both ANAC019 and Rph7 there is a large surface patch of lysine residue that are believed to be involved in binding the phosphate backbone of the DNA target. Though the major conserved residue R88 involved in DNA binding in ANAC019 and conserved in other NAC proteins is not present in Rph7. We have included this information as follows: *“Of particular importance in ANAC019 is a single, highly conserved Arg residue (Arg88) shown via mutational analysis to be required for DNA binding^{36,37}. Similarly, Rph7 shows a positive surface indicating that like ANAC019 it may be capable of binding DNA (Extended Data Fig. 4A, B), though Rph7 lacks the conserved Arg³⁵.”*

Would dimerization occur? One could try and model the dimer or see if dimerization interface is reasonable

We have previously included mention of dimerization in Rph7. Both residues reported in ANAC019 are conserved in the sequence of Rph7 and further are located in a similar region on the predicted structure as compared to ANAC019 (see Figure 2C insert). We have extended this and predicted the model of Rph7 with AlphaFold multimer (as well as just the NAC-like domain) but both models produced had a low confidence score. We have added updated and added to our previous sentence and this now reads as follows: *“Dimerization in ANAC019 is largely mediated by two prominent salt bridges formed by conserved R19 and E26 residues, which are conserved in Rph7 sequence and additionally localise in a similar region on the predicted Rph7 model (Fig. 2C). We predicted a putative dimeric Rph7 using AlphaFold2-multimer³⁶, though the interface pTM score (ipTM) was <0.2 indicating low model confidence that did not represent the experimentally determined dimeric ANAC019 structures.”*

Why do the authors assume a flexible linker? Just based on poor AF2 score? It needs to be explained

We have updated the previous sentence with respect to the flexible linker as we presumed based on the poor pLDDT score that this would likely be a region of disorder/flexibility but the reviewer is correct that this may not be the case. We have updated the sentence to *“D209 occurs within the poorly predicted region of the AlphaFold2 model between the NAC and BED domains.”*

7) Ideally hypotheses about the molecular structure and function should be confirmed by expressing the protein *in vitro* or making *in vivo* mutants based on the structure. I do accept though that this is outside the main scope of the work.

I definitely agree with the main conclusions that this is a NAC transcription factor, possibly with a Zn finger attached. However all functional properties of the two domains would need to be verified. If production of purified protein is impossible, perhaps *in vivo* mutagenesis guided by the structure could be attempted, but I do not know if this is realistic.

We agree with the reviewer that to confirm the hypothesised function of Rph7 further *in vitro* or *in vivo* studies should be undertaken, and this is a focus of our ongoing work in this project. Here, our predominant focus is on the cloning of *Rph7* and not the full functional characterisation of *Rph7*. We ensure throughout the text we use words such as “putative” or “hypothesise” with respect to a function for *Rph7* to ensure it is clear that conclusions drawn are based solely on structural predictions and comparisons and have not been validated experimentally.

Reviewer #3 (Remarks to the Author):

The study identified a barley leaf rust resistance gene Rph7. Briefly, fine mapping was performed to an interval with five genes. RNA-Seq was generated for resistance and susceptible lines with and without the treatment of multiple pathotypes. Mutagenesis and stable transformation were used to confirm the gene that encodes a protein with a putative NAC domain and a zinc finger BED domain.

The study provided solid evidence to support the cloning of the resistance gene. No major concern about this study although I feel that the writing of the manuscript could be improved. For example, when describing expression regulation, it would be helpful to be clear on what comparison was referred to. Some minor comments are listed below.

1. It was mentioned that Rph7 is a semi-dominant allele. However, transgenic lines seem to become a dominant allele (P128: 3 resistance : 1 susceptibility). Additional comment about the phenotypic difference between the native allele and the transgenic allele will be appreciated.

We thank the reviewer for this valid observation, the transgenics were phenotyped using a qualitative (compatible vs incompatible infection type) rather than a quantitative phenotype therefore the question as to whether the Rph7 allele is semi dominant in the transgenic families can not be answered. A footnote under Supplementary Table 5 has been added to capture this.

2. The Abstract is poorly written.

1) The following sentence includes rich information but is unclear: "We identified three up-regulated and pathogen-induced genes with presence/absence variation (PAV) at this locus that were only expressed in response to Rph7 avirulent pathotypes of *P. hordei*."

2) "Progeny from four independent transgenic lines segregated for the expected avirulent Rph7 infection type in response to several avirulent *P. hordei* pathotypes, however, all plants were susceptible to a single virulent pathotype confirming the specificity." could be reworded to clarify the statement.

3) It would be helpful to add a note about ANAC019 to indicate what it is.

We adopted all suggestions by the reviewer and have edited the three sentences accordingly.

3. P107: progeny-tested eight susceptible knockout M4 families

It would be informative to describe the EMS screening result. Some description in the Methods (e.g., Ten M2 families were identified as putative Rph7 knockout mutants.) could be moved to here.

We adopted the reviewer's suggestion and modified the text accordingly.

4. P115: Description of MutChromSeq can help the audience understand the method.

We added text to describe the method.

"MutChromSeq is an unbiased approach in complexity reduction used to rapidly isolate plant genes and regulatory DNA sequences that is not reliant on recombination-based genetic mapping."

5. P141: "...three distinct haplotypic groups: H1-H3. each varying in gene content corroborating previous haplotypic data reported by Scherrer, et al. 29." This sentence needs a correction.

We have split the sentence in two and have made the correction required in the text.

6. P209: Expression of four groups of genes (JR, PR, WRKY, and NLR) was focused. But no description about WRKY or NLR in the main text. It would be interesting to understand how the responses of NLR.

As the reviewer mentions we focused on these four categories of genes as marker genes for disease response that are well documented in plants. We queried our set of detected differentially expressed genes at day 6 after infection between the near isogenic line BW758 and its susceptible recurrent parent Bowman for members belonging to this gene classes. The entire list is provided in Extended Data Fig. 7. The authors have altered the text in consideration of the reviewer's suggestion in italics below:

"Furthermore, DEGs were identified by comparing BW758 and Bowman six days after infection. Four gene classes previously associated with host defence in response to fungal attack including: Jasmonate-related (JR), pathogenesis related (PR), WRKY transcription factors and NLR genes (Extended Data Fig. 7). This data suggests, as previously reported, that NAC TFs like Rph7 play an important role in regulating or modulating the cellular plant defence responses."

7. P296 "Genes expressed with a log2 fold change >+/-2" is not correct. If log2 is a negative value, the criterium should be less than -2.

This has been corrected to log2 fold change >2 or <-2 as suggested.

Reviewers' Comments:

Reviewer #1:

Remarks to the Author:

The authors have carefully addressed my comments and followed the suggestions.

A substantial amount of new experimental data on executor gene function was added to the manuscript.

Reviewer #2:

Remarks to the Author:

The authors have competently addressed all my queries and comments in their response

Reviewer #3:

Remarks to the Author:

My previous comments have been well addressed. I went through the manuscript again and focused on figures. Here are a few comments for authors' reference:

1. In Figure 3, the panel A shows the number of DEGs for four comparisons. It would be more informative if comparisons are described in the Venn diagram.
2. In Figure 4, gene and species (specifically genus) names need to be italic. Rph7_YEP was driven by either native and Ubiquitin promoters. Although it is not critical, promoters can be added to the Figure 4A for the clarification.
3. To respond to Reviewer 1, the following sentence was added (Line 255) "Nevertheless, the expression profile data for Rph7 we present suggests that it only triggers and enhances the immune response in barley when challenged with *P. hordei* pathotypes that likely all carry the same avirulence gene encoding a secreted a transcriptional activator-like effector (TALE)." The underlying logic in this statement is not very clear to me. In addition, TALE is a term to refer a specific group of effectors secreted by the bacterium (e.g., *Xanthomonas*). It is probably not reasonable to use the term.

REVIEWER COMMENTS

Reviewer #3 (Remarks to the Author):

1. In Figure 3, the panel A shows the number of DEGs for four comparisons. It would be more informative if comparisons are described in the Venn diagram.

In light of the reviewer's comments additional text has been added to the Figure 3 legend specifying the comparisons in the Venn diagram.

2. In Figure 4, gene and species (specifically genus) names need to be italic. Rph7_YEP was driven by either native and Ubiquitin promoters. Although it is not critical, promoters can be added to the Figure 4A for the clarification.

The authors have italicised gene and species names in the figure and the figure legend. The authors did not think it was necessary to modify the figure with regard to naming the promoters as they are described in the legend.

3. To respond to Reviewer 1, the following sentence was added (Line 255) "Nevertheless, the expression profile data for Rph7 we present suggests that it only triggers and enhances the immune response in barley when challenged with *P. hordei* pathotypes that likely all carry the same avirulence gene encoding a secreted a transcriptional activator-like effector (TALE)." The underlying logic in this statement is not very clear to me. In addition, TALE is a term to refer a specific group of effectors secreted by the bacterium (e.g., *Xanthomonas*). It is probably not reasonable to use the term.

The authors have modified the text in the manuscript to clarify further and have softened the conclusions made on TALEs and Rph7.